# Rat Infestation in Gauteng Province: Lived Experiences of Kathlehong Township Residents

**DOI:** 10.3390/ijerph182111280

**Published:** 2021-10-27

**Authors:** Paul Kiprono Chelule, Ayanda Mbentse

**Affiliations:** Department of Public Health, School of Healthcare Sciences, Sefako Makgatho Health Sciences University, Pretoria 0208, South Africa; iyandambentse@yahoo.com

**Keywords:** rats, infestation, township, environment, waste management

## Abstract

Background: Rat infestation is a major public health issue globally, and particularly in poor urban communities in South Africa. Rats pose significant threats to residents in the form of disease spread and sustained physical injuries. The dearth of knowledge about the experiences of affected residents may curtail the initiation of rat control programs. This study aimed to explore the lived experiences of rat infestation among residents of Katlehong Township in Gauteng Province. Methods: This was a qualitative research study where data were gathered from selected community participants from Katlehong Township in Gauteng Province. A semi-structured interview guide was used to collect data through in-depth interviews. The interviews were digitally recorded and transcribed verbatim, and thematic data analysis was conducted using NVivo12 data processing software. The data were presented in themes and quotations that reflect the views of the participants. Results: Exactly 20 community members aged between 18 and 56 years participated in the study, 80% being females. Over half of the participants were unemployed (60%), inclusive of students. Majority of the participants were either Zulu or Xhosa speakers. Several themes emerged from the data, which included the residents’ experiences of rat infestation, troublesome rats, dirty rats, reasons for rat infestation, and sustained physical injuries. Participants intimated that waste in the environment and overcrowding in homes contributed to rodent infestation. Conclusion: Rat infestation remains a problem that causes severe distress among the residents of Katlehong Township. The experiences reported varied from psychological trauma to bite injuries and destruction of household property. Effective rodent control strategies need to be put in place to manage both the physical and mental risks posed by rat infestation in socially underprivileged communities.

## 1. Introduction

Rodent infestation is a public health problem in poor urban communities in South Africa [1]. Rodents, in the context of this research, are rats and mice. There are more than thirty communicable bacterial and viral diseases that can be transmitted by rodents, including leptospirosis, plague, listeriosis, salmonellosis, haunta virus, and pulmonary syndrome [2]. Recently, for example, rodents, particularly rats, have been reported to harbour fleas that are vectors of plague—a deadly bacterial disease—and viruses within the African continent [3,4]. Some of the diseases can be transmitted to humans through cuts, scratches, and wounds, as well as via the eyes and mouth, to people who may be in contact with contaminated water or food [5]. Furthermore, rodents are occupational health hazards, as they sometimes bite fixtures such as gas pipes or electrical wires, which may result in human injuries and disasters [6]. They are also able to contaminate food through their fur and excreta. It has also been reported that rodent infestation can lead to psychological trauma in the affected communities [7,8].

By 2030, the world’s urban population is projected to increase drastically, by 2.1 billion, and since the current dwellings will not be able to cope with the massive growth, this may create a suitable breeding environment for rats and mice [5]. Urban population growth will certainly favour commensal rodents, particularly in those areas that are least able to cope, such as informal settlements without adequate sewage disposal, sanitation, housing, or infrastructure. Poor storage of food and overcrowding in the dwellings stimulate rodent breeding [9].

The closeness of rodents to humans and the high masses of rats would be anticipated to lead, not just to nuisance, but also to potential disease epidemics. In addition to the intensification of rodent infestation coupled with the risk of contracting infectious diseases, control measures remain a global public health worry as they receive little attention and, oftentimes, infestations are left to individuals to manage [5]. Although measures to decrease rodent infestation in communities in South Africa and China have been initiated, the problem still persists [1,2]. Therefore, the rodent infestation scourge needs to be prevented through satisfactory rodent control actions. Currently, in South Africa, there is a paucity of literature on the scale of rodent infestation in the affected communities. The lack of knowledge about the experiences of affected residents may dampen the initiation of rodent control programs. Thus, this study aimed to explore the lived experiences of rodent infestation among residents of Katlehong Township in Gauteng Province.

## 2. Materials and Methods

### 2.1. Study Design and Setting

This was a descriptive qualitative study, which aimed to conduct in-depth interviews with adults residing in Katlehong Township to gain an understanding of their lived experiences with rat infestation and control practices. The study setting encompassed households in Katlehong Township in Ekurhuleni. Katlehong is a sub-urban community, sub-divided into 52 sections. The smallest section has about 300 households, while the largest section consists of more than 1400 households. There is a high prevalence of rat infestation in all sections of Katlehong. The study selected 2 sites, based on the township having 2 heterogenous zones: one zone is very informal, with no formal structures at all and no water or sanitation facilities; the other zone is slightly formal, having mostly brick structures, electricity and water. Since rat infestation affects all areas of the township more or less equally, and the required sample size is small, the inclusion of two sites reasonably represents the population of Katlehong Township.

### 2.2. Study Population and Sampling

Adult members (18 years and above) of households in Katlehong Township were selected for the study population. The sample adult members ranged from 18 years to 56 years. The study employed a purposeful, non-probability sampling method since the answers sought from the questions were related to lived events, constructed around personal judgement about suitability [10]. Participants were selected with a purpose, with the essential aim of bringing about an understanding of the questions the researcher aimed to study [11]; through purposive sampling method, the researcher could gather information from participants that was “typical” of the context of the study. In this research, the two sections were identified as risk areas and, from each section, households were conveniently selected. The aim of this sampling was to explore the in-depth knowledge of these adults regarding the rat infestation so that insight may be gained regarding this topic.

### 2.3. Data Collection

The researcher collected the data, using an audio recording tape to record the interview, after gaining permission from the participants. Data were collected over a period of seven months from May to November 2018. The data were collected with the use of a semi-structured interview guide. The interview guide was developed in English and later translated to isiZulu and Sesotho, as these are the commonly used languages in the community. Each participant was given a questionnaire to complete to obtain their demographic data at the end of the interview.

### 2.4. Data Analysis

The researcher led the interviews using English, isiZulu and Sesotho, and recorded the interviews. Qualitative research produces data that are connected to the opinions, values and behaviours of people in a social context, through textual data obtained in the form of transcripts and observational field notes. The textual data were analysed using some variant of thematic analysis [12]. The recorded data were transcribed verbatim into English, isiZulu and Sesotho language transcripts. The IsiZulu and Sesotho transcripts were translated into English and the transcripts were read carefully by the researcher. A code list was established for coding of the data into themes. To identify key themes, a thematic analysis was conducted using NVIVO 12 software (QSR International Inc., Burlington, MA 01803, USA). The English versions of all twenty transcripts were imported into NVIVO 12 software for analysis. They were then grouped into similar concepts and contexts for interpretation. Details of the results were then discussed in a form of coded themes and sub-themes, as a representation of the experiences of households with respect to rodent infestation and waste disposal practices.

### 2.5. Ethical Considerations

Ethical clearance and approval to conduct the study was obtained from Sefako Makgatho University’s Research Ethics Committee (SMUREC/H/14/2018: PG). Permission to conduct the study in Katlehong Township was approved by the Ekurhuleni Health District Research Committee.

Written informed consent was obtained from the participants after adequate information about the study was provided by the researcher. The information given to participants clarified the aim and objectives of the study. Participants were also informed about the use of an audio recorder to capture the interview. Participants were informed that their participation in the study was voluntary and that at any given point in time, they could withdraw from further participation in the study.

## 3. Results

### 3.1. Participants’ Information

Interviews were conducted with twenty individuals aged between 18 and 56 years old, this being the target age. In total, 20 individuals participated in the study, with 80% being females and 20% being males. Participants’ ages ranged from 18 to 56 years, with 33 years being the mean age. Most participants were unemployed (45%), employed (40%) and students (15%). The details are presented in the Table 1.

### 3.2. Themes

Three major themes emerged from the data, which are the experiences of rat infestation by residents, the reasons for rat infestation, and the management of rat infestation. The details of the themes are shown in Table 2 below.

#### 3.2.1. Experiences of Rat Infestation by Residents

Participants expressed that rodent infestation in their area was devastating, as the rodents damaged their living environments after entering their houses. They always sensed their presence with fear and feelings of disgust. Several sub-themes that expressed their experience emerged from the data and are outlined below.

##### Sensing the Presence of Rats in the House

Participants expressed some of their sensory experiences related to determining the presence of a rats in their houses. In particular, they felt the rats walking on them while sleeping, they heard their footsteps, and they saw their droppings and footprints. The smell of dead rats would especially indicate to participants that they were present in the house. The different sensory experiences that indicated the presence of rats in their houses were stated by participants in the quotes below.


*“When there are rats in the house, you will hear noises, when they get into the house, they bring their smell.”*

*(32-year-old female)*



*“At night they make noises, sometimes in the bed. [Then] you feel something smelling and then you can’t sleep, because they are walking on the bed. You will see droppings, [a sign] that there were rats in the house. You will even see their footprints. You [will] see them running around outside in the yard. [You will also find] the rat faeces in the yard and on the toilet floor when you wake up in the morning. When you open the car bonnet you [will] also find faeces there.”*

*(33-year-old female)*



*“When sleeping during the night, they walk around the house, as if they are the owners. Their footsteps sound like two men walking in the house.”*

*(43-year-old female)*



*“We hear the sound of the rat when it tries to get in and then we know it is there. [Then] your shoes get eaten and some little fluffy papers around the corner or near your things; your bag, or your t-shirt with a hole. Such happenings make you know that there was a rat [around]. When you are sleeping you hear funny noises, you hear something sounding and you wonder if it was a thief or a rat, and it is always a rat.”*

*(31-year-old female)*


##### Fear of Rats

Most participants expressed that the sight of a rat invoked feelings of fear in them. They did not feel good at the sight of rats, as they were large and scary. Some expressed their constant worry for their children’s safety because the rats would likely bite them. Participants also intimated that some of their domestic animals feared these rats due to their intimidating sizes.


*“I don’t feel good because they are scary. I am scared of rats; kids are also scared of the rats. They are big and scary, since I was born, I have never seen such huge rats.”*

*(38-year-old female)*



*“I am not talking about small rats, but big ones as big as a cat. I become frightened because I am scared of rats. Actually, I am very scared of them.”*

*(26-year-old female)*


Although most of the participants expressed feelings of fear when they came across rats, some participants did not know how to react when they saw them, but they had to quickly think of ways to prevent them from entering their home and destroying things.


*“I used to be scared of them before, but as I said, I grew up with rats around me. So, I’m not scared of them anymore. When I see it, I just see a rat, but I have that [feeling in me] that I need to scare it away, so it won’t enter the house because if it does, it damages a lot of things.”*

*(29-year-old female)*



*“We live with them [and] we don’t have a problem. What annoys us is that they damage propert.Iif they were not causing any damage, we would live peacefully with them, you know.”*

*(34-year-old male)*


##### Feelings of Disgust

Although most participants had feelings of fear when they came across a rat in their household, some felt disgusted and annoyed because they knew that seeing a rat meant that damage to clothes, food and property would ensue. One participant associated his feelings of disgust with unclean places because of his knowledge that rats are dirty in their nature.


*“I feel disgusted when I see a rat, because where there are rats, [then that place] is not clean.”*

*(24-year-old male)*



*“So, when I see them, I get angry because I know that something is already damaged in the house; it’s just a matter of time for me before I check and realize that definitely something is not right.”*

*(30-year-old female)*



*“I just scream and jump, not that I’m afraid. [It is] just that it is irritating. It is really irritating [that] they are everywhere; they always cross [over] from the garage to my room. We used to put the dustbin that side [at the garage] and I would step on them [rats].”*

*(31-year-old female)*



*“I feel disgusted, [as] I am sure that when I see it, there will be damage [of household things]. When you see it, you must try and locate where it got in, and try to close the hole. You even find them in the toilet, [as] they dig a hole in the ground and come in. They can dig through cement. So here, because it is an informal settlement [with no cemented surfaces], they do as they please.”*

*(48-year-old male)*


##### The Troublesome Rats

During the interviews, participants expressed that the presence of rats in their homes had a negative impact on their lives. This theme was associated with feelings of helplessness and of worry due to the damage caused by the rats. Participants strongly felt that the rats were troublesome.


*“They are troublesome, and they are [so] fast, [that] you can’t catch them. I can say that they are troublesome, but we are used to it now. We try to fight them but there are always more”.*

*(48-year-old male)*



*“They are very troublesome, [as] they make a mess of everything in the shop and then even in the house, even the power lines and even the children”.*

*(32-yearold female)*



*“They are troublesome; they destroy anything they come across. They bother us because they eat our things.”*

*(38-year-old female)*



*“They destroy; they enter the house and eat our things like mealie-meal. They just make a lot of mess.”*

*(18-year-old female)*


##### Dirty Rats

The expression “dirty rats” was used repeatedly during the interviews, and thus, was selected as a theme. The issues surrounding this view of dirty rats included the environment, the nature of the rats being dirty, and food that the rats contaminate. Participants expressed that the nature of rodents being dirty and associated with waste makes them unwanted, and that their presence in the houses affects and contaminates the food they consume.


*“When there are rats, the place can’t be clean. It’s not safe here, because this place is not clean. They take dirt outside and bring it inside. There are places that look like squatter camps, the dumping site is closer to houses.”*

*(24-yearold male)*



*“Rats love dirty places, if we can just agree as people to keep the place clean, they won’t have a place to live. It’s the dirt that invites them. Let me also just say [that] anywhere [where] there is dirt, is an attraction for rats.”*

*(30-year-old female)*



*“Rats like dirt, when they move from the dumping sites, they enter our homes.”*

*(27-year-old female)*



*“[The] problem with the rats is that they also make our clothes dirty, because everywhere they walk is dirty.”*

*(36-year-old female)*


#### 3.2.2. Reasons for Rat Infestation

One of the themes that arose from the study comprised the reasons for infestation as it became important to ask the participants about their views on what seems to make these rats keep growing rather than decreasing in number. Several sub-themes that expressed the experience emerged from the data on the reasons behind the infestations, which included their incredibly huge numbers, the ease of their entry into the households, poor waste disposal, and frequent sewer problems. The sub-themes are further discussed below.

##### Lots of Rats in the Area

The participants’ perception was that there were lots of rats in the area. Some participants went on to describe the sizes of the rats they saw.


*“They are so many and troublesome. It’s so big and like a dog.”*

*(43-year-old female)*



*“There are a lot of rats in this area. They were re-blocking the shacks, [as] there were a lot of rats that went out.”*

*(26-year-old female)*



*“There are so many rats, it’s not even nice.”*

*(27-year-old female)*



*“There are too many rats here; I started seeing them from the time I was still growing up till now. They are still here.”*

*(29-year-old female)*


##### Poor Waste Disposal

Participants felt that food remains were not properly disposed of, and this provided rats with food that enabled them to reproduce and multiply. Their responses are depicted in the following quotes.


*“I think it is because they get food from our dustbins [that they exist]. If there was no food, they would find other places [to go] and they would not be much here. They are growing because new ones are born from time to time. We kill them, but we still have them, so we cannot say we have killed all of them and they no longer exist.”*

*(33-year-old female)*



*“They come because of dirt or food that has been thrown out in the yard. They also like watery places and they therefore, like to come where there are drains, especially close to kitchens. In those drains they normally find food particles because when we wash, maybe rice or any other food or veggies in the kitchen sink, it goes through the pipes and maybe, land on top of the drain. So, they like such places because they get food and water.”*

*(50-year-old female)*


##### Entry of Rats into Households

In addition to the frustration the participants felt with regard to the rat problem, they revealed that the rats usually entered their households through open doors. Participants expressed that the rats would also dig holes and enter through the roofs of their homes.


*“In my house they once entered the roof and stayed in the ceiling, we couldn’t sleep at night because they were running all over the roof and making funny noises like cats. It only got better when I found someone to come and close off the place that had a hole in the roof. Here in the garage, sometimes the door would not be properly closed, and the small rats will enter. Those ones are very problematic [because] you just see them running around. They used to come in numbers into my roof at night. They would run around in the roof.”*

*(50-year-old female)*



*“They create holes on our doors, and the other problem is that you do not see how they come in. You just discover that they already inside the house. I think that any small space that it finds, it creates a way to enter [through it].”*

*(34-year-old male)*


##### Frequency of Sewer Problems

Participants reported that there were sewer problems in the area and that the frequency of the problem varied from section to section of the township. Some participants reported that the problem would last for about a week, but eventually the municipality would come to resolve the problem if it was reported.


*“Sometimes it lasts up to 1 week. I don’t know whether the problem of sewage blockage been reported or not.”*

*(38-year-old female)*



*“It leaks and after some time it stops. I think there is somebody who reports the sewage problems.”*

*(43-year-old female)*



*“Yes, it happens frequently, almost every month. There are always blockages. We do not know where to go or who to report it to, so [that] it can be fixed.”*

*(21-year-old female)*



*“We call the municipality. It takes days. They do come, but after a long time.”*

*(24-year-old male)*


#### 3.2.3. Management of Rat Infestation

Given the experience of waste disposal problem in the area, the participants were asked to discuss their waste disposal practices. Several sub-themes emerged from the interview, which included barriers to prevent the entry of rats into houses, the removal of residential waste, and the handling of sewer problems. The sub-themes are further discussed below.

##### Barriers Created to Prevent Rats from Entering the House

One of the questions asked during the interview was how the participants prevented the rats from coming into their houses. The participants created barriers to prevent rats from coming into their homes, seeing that there was no other way but to kill these rats that had annoyed them. Some participants resorted to using pesticides, which they felt would work, although some expressed fear of using them because they feared poisoning their children. Some participants used alternative strategies to prevent rat entry, rather than only relying on pesticides. For example, some participants resorted to setting up traps and using poison simultaneously. It is important to also note that these barriers were only helpful in catching the small rats. Catching the larger rats seemed to be a challenge to most participants.


*“I use the black rat poison [which] I mix it with food, and I put that food where I know they will come and eat it, then they die. I then smell the bad odour of the dead rat inside the house. They are a problem because sometimes they die in a place where you cannot even reach easily. You have to search the whole house to find it. So, they are a problem dead or alive, but I do kill them.”*

*(33-year-old female)*



*“Most of the time we use glue that’s yellow inside. We open it [and] smear it somewhere and put food [as a bait] for them to come and eat. That is what helps us, but we cannot catch the big ones. There is nothing I can do, except to buy things that will chase them out, like the glue, rattex and aliphirimi. But because I have kids in the house, I prefer glue because it is safer than the others.”*

*(30-year-old female)*



*“The door needs to always be closed and on top of that, I need to put an old rug underneath to prevent them from coming in. It is not easy because [we need to make sure that] the poison that we use to kill them, is not easily accessible to children.”*
*(36-year-old female*)


*“I bought the glue to put inside the house, to catch the small one that would sometimes come into the house. I sprinkled some chilies all over and after doing that I never saw the rats there again. Chilies is number one [rat repellent] though, because you can also use it in cars that are parked in the yard. You can sprinkle it in the engine area and anywhere else the rats could enter. They will never go close to the car again.”*

*(50-year-old female)*


##### Strategies Used to Patch up the Excavated Holes by Rats

After participants realized that the rats had been in their houses and caused some damage, they could not merely sit and do nothing; they felt that they had to try patch up the holes that were dug by rats for entry into the houses.


*“I broke the bottles and [used them] to fill the holes dug by the rats.”*

*(38-year-old female)*



*“I sweep the sand, fill the holes with stones.”*

*(43-year-old female)*



*“I have to get rid of the soil and replace it with rocks so that the hole is closed, because the rat can dig where there is cement.”*

*(24-year-old male)*



*“Other people say that when they dig holes, you need to crush bottles and put them into those holes, and then they will not come again.”*

*(50-year-old female)*


##### Removal of Waste

Most participants alluded to the fact that the municipality came at least once a week to collect waste around the area. However, those that resided in the informal settlements did not benefit from the municipal services. When it happened that the municipality did not come to collect the waste, some participants resorted to other methods of managing their waste, such as dumping them in no-dumping zones, while others felt that they could not do anything but wait for the municipality to collect their waste.


*“Yes, they do come to collect waste. If they did not come, I just leave the waste inside the dustbin. [For] the one that is created when I do gardening, I throw [it] in an open space in my area.”*

*(34-year-old male)*



*“There is no other place where we can dispose the waste, except to go to the no-dumping areas. That causes dirt to increase and then rats will also increase there.”*

*(18-year-old female)*


##### Response to Sewer Problems

After establishing that the residents did have sewer problems in the area, the question that was then asked was whether the residents reported the problem to the municipality, in addition to ascertaining the timeframe within which the municipality would respond. Participants responded with the following:


*“If you report the problem, it might take 2 days maybe for them to send someone to fix it.”*

*(33-year-old female)*



*“The last time we had a problem, their service was poor, but they did eventually come. They came after many of us had to call and report the blockage, so we had many references. But before then, they used to come after being called by one person.”*

*(50-year-old female)*



*“Yes, we do report, and they come with the truck to fix it”.*

*(29-year-old female)*



*“I do report it because I am part of the committee for the infrastructure. Yesterday, I went to the office to ask them to open the water, because there was no water here. If someone reports sewage problems, they do come and help. They respond quickly as long as it is reported.”*

*(48-year-old male)*


## 4. Discussion

This study investigated experiences of rat infestation in an informal sub-urban community of Katlehong Township in Gauteng Province. Although we did not precisely identify the specific type of rats involved in the infestation, the most common species reported in similar environments include brown rats (*Rattus norvegicus*) and black rats (*Rattus rattus*) [13]. However, based on the participants’ descriptions of the rats as being large, as well as their sizes being comparable to dogs, and the scale of the burrows they made into the buildings, the brown rat is the most consistent fit [14]. The study participants were mainly young females in the 18–35-year category, and they were mostly unemployed and living in small households of mainly 2–4 members. The respondents came from the major ethnic groups in South Africa, Xhosa and Zulu. Katlehong Township is a high-density population area in Gauteng with poor community development and a high level of unemployment. The area is overwhelmingly crowded, with tiny yards. Facilities are lacking, with water shortages and poor sanitation. For example, it is common for around six family members to share one toilet. The conditions of living in underdeveloped environments and earning low or no income have previously been identified as contributory factors to rodent infestation [7,15]. Furthermore, living in a low socio-economic environment with limited resources alone invokes feelings of anger, powerlessness, and depression [16].

In this study, the complaints comprised both environmental and social issues. Regarding the environmental issues, for example, the participants knew that the rats’ numbers were increasing at an alarming rate in their township and believed that their area was infested because it had too much litter, which served as rat shelters, and that the infestation was also facilitated by the improperly disposed food remains that rats would feed on. The results showed that overcrowding in the homes and the deteriorating conditions of households had a high impact on rat infestation. This finding is corroborated by another study conducted in Johannesburg, which reported an association between the increased prevalence of rodents and the area of residence [17]. Some participants believed that the inadequate structure of the informal settlements contributed to the increase in rats in the area. For example, they perceived that the open spaces and dump sites that were used for their domestic waste disposal contributed to rat infestation. These findings are not unique to this community as the literature reports that rising urbanisation, human behaviour, and environmental deterioration have supported the rapid breeding of rodents [14,15]. Additionally, one United Kingdom study identifies higher prevalence of both rats and mice in dwellings where pets or livestock are kept in the garden [18]. In many cases, such dwellings would have sewer problems, unkempt spaces, unclean water, and poor sanitation [19].

The study participants reported that rats were troublesome as they dug holes in the yard, spoiled food, and damaged clothes. Occasionally, participants also experienced rat bites. Their perceptions align with findings from prior studies reporting on threat posed by rats to human health through their bites, disease spreading, and damage to food stocks and household items [20,21,22]. Similarly, bite injuries inflicted by rats often happen inside of residential houses in South Africa [23].

This study also reported extremely emotional experiences of residents during their encounters with rats in their township. For most participants, seeing a rat invoked feelings of fear because of the size of the rat and the fur on their skin. They further expressed their disgust by screaming and jumping upon encountering a rat. It is interesting to also note that other participants had feelings of disgust and irritability that were more pervasive than the fear of seeing a rat, as demonstrated in their facial expressions. Disgust is an extreme emotion that is demonstrable in humans’ feelings towards such items as faeces, vomit, wounds, corpses, rotting meat, lice, and worms. It is believed that the facial expression of disgust is universal, familiar, and cuts across cultures [24].

In this study, the participants felt that rats were dirty pests. The phrase “dirty rats” went together with the dislike of rodents and the perception that they were carriers of diseases. This view is shared by other authors, who believe that the sighting of a rat signifies dirty place because they would build nests anywhere, urinate and defaecate within that habitation [8,15]. Often, the dirt is created by the residents, when they dump waste in their yards, creating an ideal habitat for the rats [13].

In addition to residents’ experiences of rat encounters, the participants believed that their senses were a clear indicator that rats were in their households. Specifically, participants reported on feeling, hearing, and seeing the rat’s faeces and droppings in their households. In other studies, where households were surveyed, residents saw rats or their faecal droppings, which evidently proved rodent infestation [2,25]. Now that residents felt the presence of rats in their households, it was important to understand, from their perspective, how they gained entry. They knew that rats had specific means of gaining entry, including the digging of holes under the walls, through the ceiling, and via open windows and doors. Having rats enter their homes made participants feel that their privacy was invaded. Knowing that they had come to destroy items worsened their feelings. This view was expressed in what was reported in one study regarding the characteristics of a rat: that when it is not gnawing or feeding on trash, the brown rat digs [26]. Thus, the main rat entry strategy involves digging holes into the buildings and then making nests indoors. The age, structure, and condition of the house may also determine how easy it is for the rodents to enter [9].

In this study, participants were desperate to control the rat infestation in their homes and environment. For example, some of the participants relied on non-toxic techniques such as the use of broken bottles to patch up the rat holes. This is in line with other studies advocating for non-toxic preventive methods, where broken glass is added into concrete and used to seal the patched holes before the material hardens. This effectively discouraged rodents from tunnelling through into the households [27]. The use of rat trap devices was a better alternative, as it protected children from consuming the pesticides that other participants resorted to using. Thus, the use of non-chemical methods to control rats is recommended, because it reduces the chances of secondary poisoning of animals and children.

Although it has been accepted that pesticides and rat trap devices are useful as measures to decrease rodent infestation in South Africa and China 1,2], the problem of rat infestation persists due to environmental conditions such as uncontrolled waste, inadequate sewage, and poor sanitation. In this current study, participants complained that the municipality was unhelpful in collecting waste in their communities, and that their services were inconsistent. Thus, the Kathlehong community resorted to other means of handling domestic waste including burning waste goods as well as throwing them into open spaces surrounding the households. Although the practice of burning of waste in communities is common, it generates air pollutants, leading to respiratory health symptoms in the population [28]. Perceptions that the municipality alone is responsible for waste management should be discouraged as the community should actively participate in the management of domestic waste [22]. Moreover, in this study, the participants had frequent sewer blockage problems, which the frequency varying from section to section of the township. This was possible because with the pervasive overcrowding and the old age of sewer systems, the disposal of solids into the drainage systems could have caused the bursting of sewer pipes and the overflow of sewage [29].

## 5. Limitations of the Study

This was a qualitative study where purposeful sampling and a limited number of interviews were conducted. By using this sampling approach, the researcher acquired information-rich data from cases involving the lived experiences of rat encounters within communities. Although the logic of this sampling strategy adequately addressed the study objectives, the study could not determine the associations between variables such as gender and experiences. This would require a probability sampling technique applied in quantitative studies.

## 6. Conclusions

The study sought to explore the experiences of households regarding rodent infestation and the residential waste disposal practices used by residents. The study showed that rat infestation posed both physical and mental health risks to the community of Katlehong Township. There is a need for all-inclusive mitigation strategies to address both issues, particularly in socially underprivileged communities.

## Figures and Tables

**Table 1 ijerph-18-11280-t001:** Demographic profile of study participants (N = 20).

Variable	Frequency	Percentage
Gender		
Female	16	80
Male	4	20
Age category (years)		
18–35	13	65
36–49	5	25
>50	2	10
Employment status		
Employed	8	40
Unemployed	9	15
Student	3	45
Number of members in household		
2–4	11	55
5–7	8	40
8–10	1	5
Home language		
IsiXhosa	10	50
IsiZulu	6	30
SeSotho	1	5
SeTswana	1	5
XiTsonga	2	10

**Table 2 ijerph-18-11280-t002:** Themes and sub-themes.

Themes	Sub-Themes
Experiences of rat infestation by residents	Sensing the presence of rats in the house
	Fear of rats
	Feelings of disgust
	The troublesome rats
	Dirty rats
Reasons for rat infestation	Lots of rats in the area
	Poor waste disposal
	Entry of rats in households
	Frequency of sewer problems
Management of rat infestation	Barriers created to prevent rats entering the house
	Strategies used to patch up the excavated holes by rats
	Removal of waste
	Response to sewer problems

## Data Availability

Data available on request.

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
