# Peer review of "Rat Infestation in Gauteng Province: Lived Experiences of Kathlehong Township Residents"

_ijerph, 2021, doi:10.3390/ijerph182111280_

Round 1

Reviewer 1 Report

The information contained in the manuscript is interesting because is focused over the people who is affected in many aspects by the presences of the rats. 

Is not clear why just 20 people participate in the study.

Author Response

Is not clear why just 20 people participate in the study.

This is a qualitative study, which uses purposive sampling to recruit participants.  Qualitative inquiry typically focuses in depth on relatively small samples, even single cases (n = 1), selected purposefully. The purpose of purposeful sampling is to select information-rich cases whose study will illuminate the questions under study (Patton, 1990). Thus sample size is guided by data saturation. In our case, data saturation was reached at participant 20. Basically, there is no need to continue to collect data after saturation. Qualitative methods are different from quantitative methods which typically depend on larger samples selected randomly for population representation and reduction of bias.  

Patton, M. (1990). Qualitative evaluation and research methods (pp. 169-186). Beverly Hills, CA: Sage.

Reviewer 2 Report

  • The authors sampled 2 out of 52 locations in Katlehong. A statement is needed to justify why only 2 location were sampled and not more.
  • There is no mention of the ethical clearance for conducting this study
  • 20 participants took part in the study and most were females. Majority of the findings for the fear of rats is expected as females fear rats more than males. More male participants are needed to draw more sound conclusions
  • A map with the sampled sites is required in the materials and methods
  • The study conducted was a descriptive qualitative study and as such, it cannot test or verify the research problem statistically. Secondly, there is bias due to the absence of statistical tests and the margin of error that might be observed.
  • Only in the discussion section the authors mention that the study was conducted in an informal settlement setting. This should be mentioned early in the beginning of the article 

Under references

  • Please remove the semi colons in front of the 1st reference 
  • Reference number 8 is not in text
  • Richards et al., 2015 is incorrectly spelled as Richard in the references 

Author Response

The authors sampled 2 out of 52 locations in Katlehong. A statement is needed to justify why only 2 location were sampled and not more.

The study two study sites were conveniently selected based on the Township having 2 heterogenous zones:  one zone is very informal with less formal structures at all and no water or sanitation facilities, while the other zone is more formal with brick structures, electricity and water. Since rat infestation affects all areas/zones of the township, more or less uniformly,  and that the sample size  required is small, two sites reasonably represents the whole Katlehong Township population.

There is no mention of the ethical clearance for conducting this study

This information has been added to the manuscript.

20 participants took part in the study and most were females. Majority of the findings for the fear of rats is expected as females fear rats more than males. More male participants are needed to draw more sound conclusions.

This is a qualitative study, where study findings are not based on numbers but rather words. Thus, demographic information may not sway the study findings in any way. Furthermore, we are not assessing the differences in gender responses (nor any perceived variable association), given that the sample size is small and is not suitable for that analysis (as in quantitative studies).

A map with the sampled sites is required in the materials and methods

We feel that the map may not be necessary as the motivation for choosing the 2 out of 52 sites has been given. However, this can be provided if there is a feeling that it is absolutely necessary.

The study conducted was a descriptive qualitative study and as such, it cannot test or verify the research problem statistically. Secondly, there is bias due to the absence of statistical tests and the margin of error that might be observed.

This is true because this is a qualitative study, which uses purposive sampling to recruit participants. Thus, it cannot test or verify the research problem statistically as required in quantitative studies. Qualitative inquiry typically focuses in depth on relatively small samples, even single cases (n = 1), selected purposefully. The purpose of purposeful sampling is to select information-rich cases whose study will illuminate the questions under study (Patton, 1990). Qualitative methods are different from Quantitative methods which typically depend on larger samples selected randomly to address population representation and thus avoid bias.  

Patton, M. (1990). Qualitative evaluation and research methods (pp. 169-186). Beverly Hills, CA: Sage.

 Only in the discussion section the authors mention that the study was conducted in an informal settlement setting. This should be mentioned early in the beginning of the article 

This has been addressed in methodology section, under study setting.

Under references

Please remove the semi colons in front of the 1st reference 

Done.

Reference number 8 is not in text

Reference 8 is in-text under section 2.4, data analysis. Please check.

Richards et al., 2015 is incorrectly spelled as Richard in the references 

In-text citation corrected to read “Richard et al., 2015”.

Reviewer 3 Report

This is an interesting paper.  

Methods are sometimes referred to as unstructured, other times, semi-structured - consistency is required.

Sentences: 221, 283, 290, 363, 404, 409, 412, 413, 414, 426, 428, 436, 443 should be revised to improve English & for more appropriate use of academic language.

Details about ethical approval for this study are required along with an indication of the information provided to potential participants & how consent was obtained.

More detail about the setting of the study, such as type of housing, nature of sewage and waste disposal infrastructure and service provision would help to develop the context.

Author Response

Methods are sometimes referred to as unstructured, other times, semi-structured - consistency is required.

Manuscript has been revised to be more consistent, covering the issues raised.

Sentences: 221, 283, 290, 363, 404, 409, 412, 413, 414, 426, 428, 436, 443 should be revised to improve English & for more appropriate use of academic language.

Thanks for these  useful and relevant comments. Authors have revised the English in the whole manuscript. While the recommendations are noted and welcome, words by the study participants may not be altered so much, as this may affect the credibility of the study data. They require verbatim transcription/translation from the home language. However, we have tried to improve the English as much as we can using tracked changes in the manuscript.

Details about ethical approval for this study are required along with an indication of the information provided to potential participants & how consent was obtained.

This information has been added to the manuscript as section 2.4.

More detail about the setting of the study, such as type of housing, nature of sewage and waste disposal infrastructure and service provision would help to develop the context.

This information has been added to the manuscript under the sub-heading: study setting.

Round 2

Reviewer 2 Report

Minor edit for Gauteng province. It should be written as Gauteng Province  

Author Response

Minor edit for Gauteng province. It should be written as Gauteng Province 

Gauteng province changed to Gauteng Province.

Other minor errors like spellings also corrected.